# Heuristic Algorithms for the Fair Max-Min Diversity Problem

**He ZHENG**

Department of Statistics and Operations Research
University of València
Dr. Moliner, 50
Burjassot, València. 46100
`hez2@alumni.uv.es`

**Anna Martínez-Gavara**

Department of Statistics and Operations Research
University of València
Dr. Moliner, 50
Burjassot, València. 46100
`gavara@uv.es`

**Rafael Martí**

Department of Statistics and Operations Research
University of València
Dr. Moliner, 50
Burjassot, València. 46100
`rmarti@uv.es`

## Abstract

This paper investigates the Fair Max-min Diversity Problem (FMMD), which seeks to select a subset of elements that maximizes the minimum pairwise distance while ensuring fair representation across predefined groups. We formulate the problem mathematically and propose heuristic approaches to efficiently obtain high-quality solutions. Specifically, we compare the performance of the Greedy Randomized Adaptive Search Procedure (GRASP) and the Probabilistic Tabu Search (PTS). GRASP constructs solutions with adaptive randomness and refines them through local search, while PTS integrates probabilistic mechanisms within Tabu search to enhance diversification and intensification. Experimental evaluations on synthetic and real-world datasets demonstrate the effectiveness of both approaches, with PTS consistently achieving superior performance in terms of solution quality and computational efficiency.

## 1 Introduction

Consider a set $V$ containing $n$ elements and a matrix $D = (d_{ij})_{ij}$ that records the distances between these elements. The max-min diversity problem focuses on selecting a subset $U$ of $k$ elements from $V$, with the goal of maximizing the minimum distance between any pair of distinct elements within $U$. While this diversity problem has been studied in various works, a comprehensive overview of its models and solution approaches can be found in Martí and Martínez-Gavara (2023). This optimization problem finds application in a variety of areas, such as data summarization and recommendation systems. However, in many real-world scenarios, the elements in $V$ can be divided into $C$ disjoint

groups $V_1, V_2, ...V_C$ based on specific attributes like gender or education level. In such cases, solving the standard max-min diversity problem without additional constraints may result in under- or over-representation of certain groups. To address this issue, researchers have introduced the concept of fairness constraints, giving rise to the fair max-min diversity problem (FMMD) (Moumoulidou et al., 2020; Addanki et al., 2022; Wang et al., 2023). The FMMD generalizes the original problem by maintaining the core objective, identifying $k$ elements that maximize the minimum pairwise distance within the chosen subset, while ensuring that the number of selected elements from group $V_c$, where $c \in [1, 2, ..., C]$, must fall within a predetermined range, i.e. $[l_c, p_c]$. In this context, the overall selected subset $U$ can be represented as the union of subsets $U_c$, where each $U_c \subseteq V_c$ and the cardinality of each subset $|U_c|$ must satisfy the fairness constraints.

The fair max-min diversity problem has proved to be NP-complete in Moumoulidou et al. (2020). Existing studies primarily focus on approximation algorithms, which provide theoretical guarantees on solution quality and have demonstrated remarkable computational efficiency, particularly in large-scale and streaming data settings (Wang et al., 2023). However, these methods may sacrifice solution quality for speed, which may limit their applicability in scenarios requiring high-quality solutions.

In this work, we establish a mathematical model for the problem and then investigate heuristic approaches to obtain high-quality solutions efficiently. Specifically, we compare two heuristic methods for solving the FMMD problem: the Greedy Randomized Adaptive Search Procedure (GRASP), which integrates randomness during solution construction, and the Probabilistic Tabu Search (PTS), which incorporates probabilistic mechanisms within its search. Through this comparative analysis, we aim to provide insights into the impact of different probabilistic strategies on both solution quality and computational efficiency.

## 2 Mathematical model

The problem can be mathematically formulated in the following way, where $x_i = 1$ if element $i$ is selected in $U$, and $x_i = 0$ otherwise:

$$
\begin{aligned}
\max \quad & f(U) = \min_{1 \le i < j \le n} d_{ij} x_i x_j \\
\text{s.t.} \quad & \sum_{i=1}^{n} x_i = k \\
& l_c \le \sum_{i \in V_c} x_i \le p_c, \quad \forall c = 1, 2, \ldots, C \\
& x_i \in \{0, 1\}, \quad \forall i = 1, \ldots, n
\end{aligned}
\tag{1}
$$

The non-linear objective function in the above model presents significant challenges for solving the problem. To address this issue, we linearize the formulation in the following way:

$$
\begin{aligned}
\max \quad & f(U) = t \\
\text{s.t.} \quad & z_{ij} \le x_i, && \forall 1 \le i < j \le n \\
& z_{ij} \le x_j, && \forall 1 \le i < j \le n \\
& z_{ij} \le x_i + x_j - 1, && \forall 1 \le i < j \le n \\
& t \le d_{ij} z_{ij} + M(1 - z_{ij}), && \forall 1 \le i < j \le n \\
& \sum_{i=1}^{n} x_i = k \\
& l_c \le \sum_{i \in V_c} x_i \le p_c, && \forall c = 1, 2, \ldots, C \\
& x_i \in \{0, 1\}, && \forall i = 1, \ldots, n
\end{aligned}
\tag{2}
$$

where $M$ is a sufficiently large constant to ensure the constraint holds when $z_{ij} = 0$. This linearized model simplifies the problem and enables it to be efficiently tackled using standard integer programming solvers, making the FMMD problem more tractable in practice.

# 3 Solution methods for the FMMD

## 3.1 GRASP

GRASP is a multi-start search procedure proposed by Feo et al. (1994); Feo and Resende (1995). Each iteration consists of two main phases: the first phase focuses on constructing the initial solution in an iterative, randomized, greedy, and adaptive manner, followed by the use of a local search procedure to improve the solution and achieve a local optimum. This process continues until a predefined stopping criterion is met, which, in the case of FMMD, is a cutoff time.

### 3.1.1 Construction method

To generate an initial solution of reasonable quality, we adapt the GRASP methodology for the construction procedure of FMMD. This involves iteratively selecting elements based on a greedy criterion, while also introducing randomness to explore various solutions, ultimately building the solution step by step until the desired size is achieved.

Specifically, let $U = \bigcup_{c=1}^{C} U_c$ represent a partially constructed solution, and let $d_i^*$ denote the minimum distance of $i$ to the selected elements (excluding itself), which is formally expressed as:

$$d_i^* = \min_{j \in U, i \neq j} d_{ij}. \tag{3}$$

This measure is crucial because it directly influences the objective function value, denoted as $f(U)$, which is the minimum $d_i^*$ among the selected elements, as shown below:

$$f(U) = \min_{i \in U} d_i^*. \tag{4}$$

It is clear that elements with larger $d_i^*$ are more likely to contribute favorably to the overall solution quality. To facilitate the selection of such elements, we further define $max\_d^*$ as the maximum $d_i^*$ among the candidate elements. Note that due to the fairness constraint, the candidate elements do not include all unselected elements.

At the beginning, $U$ is set to $\emptyset$, and $d_i^*$ is initialized to a large enough value for all $i \in V$. The construction procedure then proceeds with $k$ consecutive steps of addition operation to build a solution. To ensure the feasibility of the solution, the candidate elements for each step are determined based on different criteria. More precisely, for the first $\sum_{c=1}^{C} l_c$ steps, the candidate elements consist of the unselected elements from each group that has fewer than $l_c$ selected elements. This phase ensures that at least $l_c$ elements are selected from $V_c$. In the final $k - \sum_{c=1}^{C} l_c$ steps, the candidate elements are the union of the unselected elements of each group whose number of selected elements is less than $p_c$.

At each iteration, all candidate elements satisfying the condition $d_i^* \geq \alpha \times max\_d^*$ are included in the Restricted Candidate List ($RCL$). The parameter $\alpha \in [0, 1]$ controls the balance between randomness and greediness in the construction process. A higher $\alpha$ value restricts the $RCL$ to only the top-performing candidates, while a lower $\alpha$ increases randomness by allowing a broader range of candidates to be considered. Once the $RCL$ is determined, element $i'$ is randomly selected from it and added to $U$.

The objective function value of the new partial solution is then updated to the minimum value of $d_{i'}^*$ and $f(U)$, where $f(U)$ denotes the objective function value before the insertion. Furthermore, for all elements $i \in V$, the $d_i^*$ value is modified to the lesser of $d_i^*$ and $d_{i'i}$ when $i \neq i'$, while remaining unchanged otherwise.

### 3.1.2 Local search procedure

After obtaining an initial solution, we refine it through a best-improvement local search strategy. In each iteration, the algorithm exhaustively explores the entire neighborhood to identify the most beneficial move. This process continues until no further improvement is possible, meaning that all potential moves would deteriorate the current objective value.

To ensure that exactly $k$ elements are selected, we utilize the $Swap(i,j)$ operator in our algorithm, where $i$ is the element to be removed and $j$ is the one to be added to the solution. For each $i \in U$, we compute the corresponding $d_i^*$ values. Given the definition of the objective function, it is evident that removing the element $i$ for which $d_i^* = f(U)$ will lead to an improved solution, as the objective function value will either increase or remain unchanged, while the number of elements attaining the minimum value $f(U)$ will decrease.

This strategy allows us to focus on the elements that most significantly limit the solution's quality, which has been proposed for related problems (Sánchez-Oro et al., 2025; Martínez-Gavara et al., 2021; Resende et al., 2010). Instead of evaluating all elements in $U$, our approach restricts the removal candidates in each iteration to a smaller subset $X$, which includes only the selected elements satisfying $d_i^* = f(U)$.

$$X = \{i \in U | d_i^* = f(U)\}. \tag{5}$$

Taking the fairness constraint into account, the set of elements that can be swapped with element $i \in X$ is determined differently based on the selection conditions of the group to which $i$ belongs. Let $\hat{c}$ represent the group of element $i$. When the number of selected elements in group $\hat{c}$ exceeds its lower bound, i.e., $|U_{\hat{c}}| > l_{\hat{c}}$, the set of possible swap candidates $Y_i$ consists of all unselected elements within group $\hat{c}$, as well as all unselected elements from other groups where the number of selected elements has not yet reached the upper bound. The set of eligible groups for swap candidates is given by the group candidate list, $GCL$, for element $i$, $GCL_i = \{\hat{c}\} \cup \{c \in C \mid c \neq \hat{c}, \ |U_c| < p_c\}$. Otherwise, when the number of selected elements in group $\hat{c}$ is equal to its lower bound, the swap candidates for $i$ are restricted to only the unselected elements within the same group $\hat{c}$. Therefore, the set $Y_i$ can be mathematically expressed as:

$$Y_i = \begin{cases} \bigcup_{c \in GCL_i} (V_c \setminus U_c) & \text{if } |U_{\hat{c}}| > l_{\hat{c}} \\ V_{\hat{c}} \setminus U_{\hat{c}} & \text{otherwise.} \end{cases} \tag{6}$$

Once the set $Y_i$ is determined, the neighborhood induced by all the potential moves involving elements from $X$ and $Y$ is denoted as:

$$N(U) = \{U' \mid U' = U \setminus \{i\} \cup \{j\}, i \in X, j \in Y_i\}. \tag{7}$$

To assess the impact of a potential $Swap(i,j)$ operation, where $i \in X$ and $j \in Y_i$, we define an evaluation function $\triangle_{i,j}$ as the difference between two values: the minimum distance from element $j$ to the solution set excluding $i$ (i.e., $U \setminus \{i\}$) and $d_i^*$. This evaluation function guides our search strategy by replacing an element $i$ with a low $d_i^*$ value with an element $j$ that maintains a greater minimum distance to the remaining solution elements. By doing so, we can potentially enhance the overall diversity of the solution.

To reduce the computational complexity in determining the evaluation value and calculating the objective value, we use a streamline calculation strategy for the FMMD. Here, we introduce a set $A_i$ for each element $i$, which stores the neighbors of element $i$ whose distance to $i$ equal to $d_i^*$. This ensures that the minimum distance from $j$ to $U \setminus \{i\}$ is recalculated only when necessary. Specifically, it is updated only if $i$ is the sole element in $U$ with $d_{ij} = d_j^*$, in other words, if $i \in A_j$ and $|A_j| = 1$. If these conditions are not met, we retain the original $d_j^*$ value. This auxiliary set $A_i$ also assists in efficiently updating the relevant data after a swap move, as illustrated in Algorithm 1. This update process includes adjusting the minimum distance $d^*$ of each element to the new solution $U'$, and once $d_i^*$ changes, $A_i$ is updated accordingly. Additionally, the objective function value $f(U')$ is recalculated, and the set $X$ is updated for the subsequent search iteration.

### 3.2 Probabilistic Tabu Search

PTS is an extension of the classical Tabu Search (TS), which introduces probabilistic mechanisms to enhance exploration and flexibility (Glover, 1989). To better understand PTS, we first review the key components of TS, including short-term memory to prevent cycling, aspiration criteria to allow promising moves, and long-term memory to guide diversification.

**Algorithm 1** Solution Update

---

**Require:** The current solution $U$, the elements to out ($i'$) and to in ($j'$) of the swap move, the arrays or sets $d^*$, $A$ and $X$
**Ensure:** A updated solution $U'$ and the updated arrays or sets.

1:  $U' \leftarrow U \setminus \{i'\} \cup \{j'\}$        ▷ *Update the solution*
2:  **for** $i \in V$ **do**        ▷ *Update $d^*$ and $A$*
3:      **if** $d_{ij'} < d_i^*$ and $i \neq j'$ **then**
4:         $d_i^* = d_{ij'}$
5:         $A_i \leftarrow \emptyset \cup \{j'\}$
6:      **else if** $d_{ij'} > d_i^*$ **then**
7:         **if** $i' \in A_i$ and $|A_i| = 1$ **then**
8:            $d_i^* \leftarrow \min\{d_{ij} \mid j \in U'\}$
9:            $A_i \leftarrow \{j \in U' \mid d_{ij} = d_i^*\}$
10:        **else if** $i' \in A_i$ and $|A_i| > 1$ **then**
11:          $A_i \leftarrow A_i \setminus \{i'\}$
12:      **else if** $d_{ij'} = d_i^*$ **then**
13:        **if** $i' \in A_i$ **then**
14:          $A_i \leftarrow A_i \setminus \{i'\} \cup \{j'\}$
15:        **else**
16:          $A_i \leftarrow A_i \cup \{j'\}$
     $f(U') \leftarrow \infty$        ▷ *Update $f(U')$ and $X$*
17:  **for** $i \in U'$ **do**
18:      **if** $f(U') > d_i^*$ **then**
19:        $f(U') = d_i^*$
20:        $X \leftarrow \emptyset \cup \{i\}$
21:      **else if** $f(U') = d_i^*$ **then**
22:        $X \leftarrow X \cup \{i\}$
     **return** $U', f(U'), d^*, A, X$

---

### 3.2.1 Key Components in TS

In both TS and PTS, short-term memory serves as a restriction to prevent recently performed moves from being immediately reversed. Specifically, when a swap move $Swap(i,j)$ is performed, its reverse move $Swap(j,i)$ is temporarily designated as forbidden (tabu) for the next $\gamma$ iterations, a period known as the tabu tenure. To track the tabu status of moves, we maintain a tabu list $H$, initialized with zeros. Upon executing $Swap(i,j)$, we update the list as $H[i][j] = iter + \gamma$, where $iter$ denotes the current iteration number. A move remains forbidden until the iteration count surpasses its stored value, encouraging exploration by preventing immediate revisits to previous solutions.

Another important component is the aspiration criterion which provides flexibility in the search by allowing tabu moves under specific conditions. If a tabu move results in a solution that improves upon the best found so far, the tabu restriction is overridden, and the move is accepted. Consequently, a move is considered eligible if it is either non-tabu or satisfies the aspiration condition, ensuring that high-quality solutions are not discarded due to short-term restrictions.

Finally, long-term memory is a type of memory designed to promote diversification by encouraging the search to explore less frequently visited regions. To achieve this, we employ a frequency-based memory structure $L$, which records the historical occurrence of swaps between elements. Specifically, let $L[i][j]$ represent the number of times elements $i$ and $j$ have been swapped. Whenever a swap $Swap(i,j)$ or $Swap(j,i)$ is performed, the corresponding entry in the frequency matrix is updated as $L[i][j] = L[i][j] + 1$.

To discourage excessive repetition of frequently occurring swaps, this frequency value is incorporated as a penalty term and assessed alongside the evaluation function. In this work, however, the penalty is only applied to non-improving moves (Chiang and Kouvelis, 1996), ensuring that promising swaps are not unnecessarily penalized. This mechanism helps balance intensification and diversification, preventing the search from being trapped in a limited region while still allowing promising moves to be accepted without additional constraints.

### 3.2.2 Main framework of Probabilistic Tabu Search

The main steps of PTS begin with an initial solution, which is typically generated randomly or through a greedy method. The algorithm then iteratively explores the solution space by selecting neighboring solutions. In each iteration, PTS evaluates the admissible moves and, instead of strictly selecting the best move, it uses probabilistic mechanisms to sometimes accept non-best moves. This allows the algorithm to escape local optima and explore more diverse regions of the solution space.

The main process of the PTS for solving the FMMD problem is outlined in Algorithm 2. The algorithm begins with a greedily constructed initial solution $U$ (line 1) and records it as the initial best solution $U^*$ (line 2). It is well established that starting with a good initial solution enhances the performance of PTS (Guan et al., 2018; Guemri et al., 2019; Wang and Zhao, 2022). The search continues until a predefined stopping criterion is met (line 3), which in this case is a fixed time limit. At each iteration, PTS identifies a swap pair $(\hat{i}, \hat{j})$ (line 4) according to the rule defined in Algorithm 3. The selected swap is then applied to update the solution (line 5), and the tabu list $H$ along with the swap frequency matrix $L$ are updated accordingly (line 6). If the new solution improves upon the best solution found so far, it is recorded as the new best solution (lines 7–8).

In Algorithm 3, a move candidate list $MCL$ is constructed by identifying all feasible swap pairs $(i, j)$ (as presented in Section 3.1.2) that are either non-tabu or satisfy the aspiration criterion (lines 1–5). A move selection strategy (Guemri et al., 2019) is then used to probabilistically choose a swap from $MCL$ (detailed in lines 6–14). Specifically, PTS first sorts the candidates in $MCL$ in descending order based on the evaluation function $\triangle_{i,j}$, incorporating the swap frequency penalty $S$ (line 6). Then, a random number $p$ is generated to determine the selection mechanism (line 7): with probability $p_1$, the best candidate is chosen (line 9); with probability $p_2$, a candidate is randomly selected from the top $\beta$-fraction of $MCL$ (lines 11–12); otherwise, a completely random selection is made from the entire candidate list (line 14).

---

**Algorithm 2** Probabilistic Tabu Search PTS$(p_1, p_2, \beta)$

---

**Require:** Graph $G(V, E)$, the selection probabilities $p_1$ and $p_2$, and a parameter $\beta$ determining the fraction of top candidates to consider.
**Ensure:** The best solution found so far.
1: $U \leftarrow$ GREEDYCONSTRUCTION()
2: $U^* \leftarrow U$                                                                                 ▷ *Record the best solution*
3: **while** $(current\_time - start\_time) < time\_limit$ **do**
4:     $(\hat{i}, \hat{j}) \leftarrow$ DETERMINEMOVE$(X, Y, p_1, p_2, \beta)$                  ▷ *See details in Algorithm 3*
5:     $U \leftarrow$ UPDATESOLUTION$(\hat{i}, \hat{j})$                              ▷ *See details in Algorithm 1*
6:     Update the tabu list $H$, and the swap frequency $L$.
7:     **if** $f(U) > f(U^*)$ **then**
8:         $U^* \leftarrow U$
    **return** $U^*$

---

## 4 Comparative study

### 4.1 Benchmark instances

To evaluate the performance of the proposed methods, we conduct experiments on a set of benchmark instances. Specifically, we use synthetic datasets generated based on the rules and Python code provided by Wang et al. (2023), varying in the number of elements $n$ and the number of groups $C$, to assess the adaptability of the algorithms across different instance scales and group structures. Additionally, we include a real-world dataset used in Wang et al. (2023) to further validate the effectiveness of our approach in practical applications. The details of these datasets are as follows:

1. The synthetic datasets consist of 80 instances, where the number of elements ranges from 100 to 10,000 (specifically, 100, 500, 1,000, and 10,000). The elements are assigned to groups uniformly at random, with the number of groups set to 2, 4, 6, and 8. For each instance, we select $k = 10, 20, 30, 40, 50$ elements.

2. The real-world instance is derived from the Adult dataset, which contains 48,842 records from the 1994 US Census database. The group number is 2 (based on sex), 5 (based on

---

**Algorithm 3** Determine the move DETERMINEMOVE($X, Y, p_1, p_2, \beta$)

---

**Require:** The removal candidates $X$, the addition candidates $Y$, the selection probabilities $p_1$ and $p_2$, and a parameter $\beta$ determining the fraction of top candidates to consider.

**Ensure:** A swap pair $(\hat{i}, \hat{j})$.

1: $MCL \leftarrow \emptyset$                                                  ▷ *Candidate list construction in PTS*
2: **for** $i \in X$ **do**
3:      **for** $j \in Y_i$ **do**
4:          **if** $Swap(i, j)$ is non-tabu or satisfies the aspiration criterion **then**
5:              $MCL \leftarrow MCL \cup (i, j)$
6: Sort $MCL$ in decreasing order based on the evaluation function $\triangle_{i,j}$, incorporating the swap frequency penalty $L$.                ▷ *Move selection mechanism*
7: Generate a random number $p = \text{random}(0, 1)$
8: **if** $p \leq p_1$ **then**
9:      Select the optimal candidate $(\hat{i}, \hat{j})$ from $MCL$.
10: **else if** $p \leq p_2$ **then**
11:      Let $b = \beta \times |MCL|$
12:      Randomly select $(\hat{i}, \hat{j})$ from the top-$b$ candidates in $MCL$.
13: **else**
14:      Randomly select $(\hat{i}, \hat{j})$ from the entire candidate list $MCL$.
      **return** $(\hat{i}, \hat{j})$

---

race), and 10 (based on the combination of sex and race). $k$ values are the same as those of the synthetic datasets, resulting in a total of 15 instances.

Given a value of $k$, the upper and lower bounds for the number of selected elements from each group, $l_c$ and $p_c$ respectively, can be set either based on proportional representation (El Halabi et al., 2020; Celis et al., 2018) or equal representation (Wang et al., 2023; Kleindessner et al., 2019; Wang et al., 2021). In this study, we adopt the latter approach, which is more commonly used when addressing fairness constraints.

## 4.2 Experimental results

The proposed heuristics are implemented in C++ and compiled using GNU GCC 10.2.0 with the -O3 optimization flag. We also run SFDM, the state-of-the-art approximation algorithm for the FMMD problem in the streaming model, using the Python 3 implementation provided by Wang et al. (2023). To ensure high solution quality, we set the approximation ratio parameter to 0.01. All experiments are conducted on a machine equipped with an Intel(R) Gold 6226R processor (2.90GHz), running the Linux 3.10.x86_64 operating system. The heuristic methods and the approximation algorithm are all applied to each instance 10 times. Specifically, for GRASP and PTS, a time limit of 30 seconds per run is imposed on instances with fewer than 500 elements (i.e., 100 and 500). For larger instances, the cutoff time is extended to 60 seconds per run. Additionally, we use the CPLEX solver to solve the linear model for this problem, with a time limit of 10 hours.

Table 1 presents a performance comparison of SFDM, CPLEX, GRASP, and PTS for solving the FMMD problem across synthetic and real-world datasets. For CPLEX, we report the number of instances solved to optimality ($\#opt$), the number of instances where CPLEX fails to find a solution within the time limit ($\#fail$), the average percentage deviation ($\%dev$) from the best solution obtained among the three methods, and the computation time required to generate a solution ($time$). For GRASP and PTS, we provide the number of instances where each method achieved the best-known solution at least once out of ten executions ($\#best$), the success rate over 10 independent runs ($\#succ$), the percentage deviation of the best solution obtained by each heuristic from the optimal or best-known value ($\%dev_b$), the percentage deviation of the average solution across all runs from the optimal or best-known value ($\%dev_a$), and the average computational time ($time$).

The results indicate that while CPLEX can solve small instances to optimally, its performance deteriorates significantly as the problem size increases. For larger and real-world datasets, CPLEX fails to find a solution due to the overwhelming number of variables and constraints. Although the state-of-the-art approximation algorithm is capable of handling real-world instances fast, it frequently fails on smaller instances, failing to provide a feasible solution for 24 instances. Even when it does

Table 1: Comparison results of SFDM, CPLEX, GRASP and PTS on synthetic and real-world datasets

| | $n$ | $\#ins$ | SFDM | | | | CPLEX (10 h) | | | | GRASP | | | | | PTS | | | | |
|---|---|---|---|---|---|---|---|---|---|---|---|---|---|---|---|---|---|---|---|---|
| | | | $\#fail$ | $\%dev_b$ | $\%dev_a$ | $time$ | $\#opt$ | $\#fail$ | $\%dev$ | $time$ | $\#best$ | $\#succ$ | $\%dev_b$ | $\%dev_a$ | $time$ | $\#best$ | $\#succ$ | $\%dev_b$ | $\%dev_a$ | $time$ |
| Synthetic datasets | 100 | 20 | 10 | 35.02 | 48.19 | 1.46 | 20 | 0 | 0 | 554.69 | 19 | 9.4/10 | 0.13 | 0.22 | 1.64 | 20 | 10/10 | 0 | 0 | 0.42 |
| | 500 | 20 | 5 | 34.09 | 44.60 | 3.03 | 3 | 0 | 23 | 32322.22 | 11 | 3.9/10 | 0.95 | 1.74 | 11.70 | 20 | 8.8/10 | 0 | 0.06 | 3.70 |
| | 1000 | 20 | 5 | 34.72 | 44.66 | 4.40 | 1 | 0 | 41.48 | 35134.13 | 6 | 3.8/10 | 2.18 | 2.86 | 22.95 | 20 | 5.4/10 | 0 | 0.31 | 13.94 |
| | 10000 | 20 | 4 | 34.53 | 42.47 | 6.00 | 0 | 20 | - | - | 1 | 2.4/10 | 3.90 | 5.35 | 28.70 | 20 | 2.5/10 | 0 | 0.86 | 29.10 |
| Real-world dataset | 48842 | 15 | 0 | 23.47 | 29.26 | 52.05 | 0 | 15 | - | - | 2 | 1.1/10 | 2.97 | 5.61 | 30.73 | 13 | 1.4/10 | 0.21 | 2.19 | 36.42 |
| $Sum$ | | 95 | 24 | | | | 24 | 35 | | | 39 | | | | | 93 | | | | |
| $Avg$ | | | | 32.37 | 41.84 | 13.39 | | | 21.49 | 22670.34 | | 4.1/10 | 2.02 | 3.16 | 19.14 | | 5.6/10 | 0.04 | 0.69 | 16.72 |

succeed, the solution quality remains poor, with a significant gap from the optimal or best-known solutions. In contrast, both GRASP and PTS provide high-quality solutions efficiently, with PTS consistently achieving the best performance in terms of solution quality and computational time. Specifically, for small instances (e.g., $n = 100$), both heuristics achieve near-optimal solutions, but as the problem size increases, GRASP struggles with local optima, whereas PTS effectively navigates the search space, yielding solutions that are on average 2% better. Additionally, PTS exhibits greater stability, with lower variance across multiple runs, suggesting its robustness in handling diverse instances.

To further illustrate the behavior of the heuristics, we present the convergence results on a set of more challenging instances, where $n = 1000$ and $k = 30$. Figure 1 contains four subgraphs, each representing the convergence behavior with 2, 4, 6, and 8 groups, respectively. The vertical axis represents the average objective value over 10 runs, while the horizontal axis indicates the running time. Each point on the graph corresponds to an update at that moment. The experimental results show that PTS converges more quickly and produces higher-quality solutions, while GRASP continues to improve its best solution in the later stages of the search. However, within the given time frame, GRASP does not achieve the solution quality of PTS. Additionally, in instances with a higher number of groups, PTS consistently delivers high-quality solutions, while the performance of GRASP shows a noticeable decline. These results suggest that PTS exhibits strong adaptability when solving the FMMD problem, maintaining stable and high-quality performance in the complex scenarios.

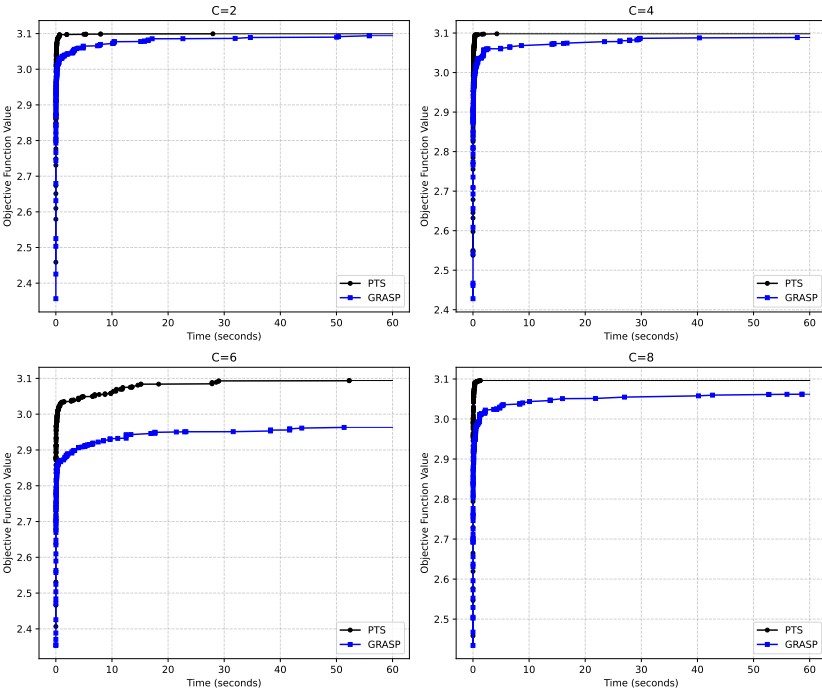

Figure 1: Convergence analysis of PTS and GRASP over time in selecting 30 elements from 1000.

# 5 Conclusion

In this study, we addressed the FMMD problem by developing and comparing two heuristic methods: GRASP and PTS. Our findings indicate that while GRASP provides a competitive baseline through its adaptive construction and local search, PTS outperforms it by exploiting memory structures and probabilistic selection mechanisms. Experimental results demonstrate that PTS consistently produces higher-quality solutions within shorter computational times, especially for larger problem instances. These findings highlight the effectiveness of probabilistic memory-based search strategies in solving fairness-constrained diversity problems.

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
