# OpenReview forum: "Heuristic Algorithms for the Fair Max-Min Diversity Problem"
_MAEB/2025/Congreso — MAEB 2025_

### Official Review · Reviewer_2j8b · 2025-03-08
**Metaheuristic search for solving the FFMD problem, with outstanding results**

**Rating:** 5
**Confidence:** 4

**Review:**

This paper studies the Fair Max-min Diversity Problem (FMMD), and proposes to solve it using two metaheuristic algorithms, namely Randomized Adaptive Search Procedure (GRASP) and Probabilistic Tabu Search (PTS). Results show that their implementation of both algorithms outperform an optimal solver (CPLEX) that cannot handle larger versions of the problem.

Issues to be addressed:

- tabu search ---> Tabu search
- in the recent book (ref) ---> in (ref)
- "may may result"
- IN the introduction, the authors should discuss potential benefit or pitfall of using a metaheursitic for this problem, compared to an approximation algorithm, and what are the state of the art algorithms and benchmark problems
- When defining FFMD, state what type of problem is it (NP, NP-Complete, NP-Hard)
- Define acronyms consistently in the main text on first use (abstract is separate, definitions there are not relevant to the main text), and do not redefine them
- Equations are not like floats (tables, figures), even if numbered, they are part of the text. They cannot be referenced before they are defined (even if they appear on the next line of text), and need to respect grammar, if the following paragraph begins with a capitalized word, then the equation should end in a period
- "We also run the state-of-the-art approximation algorithm for the FMMD problem proposed by Wang et al. (2023) on these instances. However, due to its poor result quality—even on datasets with only 100 elements—its results are not included in this study." WHY? if it has poor results and it is the state of the art, then why not contrast your results with those? seems like an odd choice to leave it out, please add them

---

### Official Review · Reviewer_VrkF · 2025-03-11
**This work elaborates on the Fair Max-Min Diversity (FMMD) problem, that extends the max-min diversity problem with additional constraints to ensure that several partitions of a set of elements are well-represented. The authors have formulated the problem as a nonlinear optimization problem, that has been linearized afterwards so to make it affordable by CPLEX. Two heuristic methods, a GRASP and a probabilistic Tabu Search (PTS) has been developed and evaluated over a set of both synthetic and real-world instances. The paper is very well written and easy to follow.**

**Rating:** 5
**Confidence:** 4

**Review:**

This work elaborates on the Fair Max-Min Diversity (FMMD) problem, that extends the max-min diversity problem with additional constraints to ensure that several partitions of a set of elements are well-represented. The authors have formulated the problem as a nonlinear optimization problem, that has been linearized afterwards so to make it affordable by CPLEX. Two heuristic methods, a GRASP and a probabilistic Tabu Search (PTS) has been developed and evaluated over a set of both synthetic and real-world instances. The paper is very well written and easy to follow. Only several minor issues are raised:

1.- In Eq (1), the definition of f(U) requires to include the domain of i and j, as in the last constraint.

2.- Using the runtime as stopping condition is controversial, as it strongly depends on the programming language used, the programming skills of the authors, the execution platform, if it can be guaranteed that the computing facilites are dedicated and no other running process interfere, etc. With a constructive heuristic, using a predefined number of function evaluations is also unfair. The authors may want to show them both.

3.- On a related matter, please elaborates a little bit more on why GRASP does not compute for almos the time limit in the largest instances, if it seldom reaches the best-known solution.

3.- The authors may want to complete up to 30 independent runs and provide the results of GRASP and PTS with statistical confidence.

---

### Official Review · Reviewer_UUZm · 2025-03-19
**This paper  proposes two heuristic algorithms (GRASP AND PTS) for the solution of the fair max-min diversity problem. PTS shows the best results.**

**Rating:** 5
**Confidence:** 4

**Review:**

The originality, organization and clarity of this paper are very good. The problem is clearly stated from the beginning, an integer-linear formulation is presented, and then the two proposed heuristic methods are also presented in detail.

The state problem is interesting and relevant, the experimental framework is well designed and includes instances with a high number of variables (up to 48882) and with a convincing comparison among the methods evaluated.

I do not have any serious criticism of the paper, just some few comments:

1) Page 2., pag. 43. It would be good to make a reference to the  number of variables in the linear formulation of the original problem (Eq. 2).

2) In the experimental framework, it is not explicitly mentioned what is the criterion used to consider the success of an algorithm. It is assumed that for the algorithms to be successful in reaching the best known solution, it has to find the best solution at least once in ten executions.

3) CPLEX can use a number of methods to augment the efficiency. For example, it is possible to reduce the number of original variables, that in the linearization of the original problem are many. Was any of these preprocessing steps applied ?

4) In Table 1, it is not clear in how many of the 10 runs the best solution was found.

5) Providing some more information about the behavior of the heuristics could help to understand which component of the algorithms plays the most relevant role (e.g., to what extent is the initialization important)

6) The two heuristics introduced work by improving single solutions. This problem could be more suitable for population-based optimizers able to combine partial solutions from different solutions, particularly is the solutions are initialized using some a-priori information or greedy heuristic.

Other comments:

Page 1, line 23: may may

---

### Decision · Program_Chairs · 2025-03-20

Accept